# Essential Oils Biofilm Modulation Activity and Machine Learning Analysis on *Pseudomonas aeruginosa* Isolates from Cystic Fibrosis Patients

**DOI:** 10.3390/microorganisms10050887

**Published:** 2022-04-24

**Authors:** Marco Artini, Rosanna Papa, Filippo Sapienza, Mijat Božović, Gianluca Vrenna, Vanessa Tuccio Guarna Assanti, Manuela Sabatino, Stefania Garzoli, Ersilia Vita Fiscarelli, Rino Ragno, Laura Selan

**Affiliations:** 1Department of Public Health and Infectious Diseases, Sapienza University, p.le Aldo Moro 5, 00185 Rome, Italy; marco.artini@uniroma1.it (M.A.); rosanna.papa@uniroma1.it (R.P.); gianluca.vrenna@uniroma1.it (G.V.); 2Rome Center for Molecular Design, Department of Drug Chemistry and Technology, Sapienza University, p.le Aldo Moro 5, 00185 Rome, Italy; filippo.sapienza@uniroma1.it (F.S.); manuela.sabatino@uniroma1.it (M.S.); 3Department of Drug Chemistry and Technology, Sapienza University, p.le Aldo Moro 5, 00185 Rome, Italy; stefania.garzoli@uniroma1.it; 4Faculty of Natural Sciences and Mathematics, University of Montenegro, Džordža Vašingtona bb, 81000 Podgorica, Montenegro; mijatboz@ucg.ac.me; 5Research Unit of Diagnostical and Management Innovations, Children’s Hospital and Institute Research Bambino Gesù, 00165 Rome, Italy; vanessa.tuccio@opbg.net (V.T.G.A.); evita.fiscarelli@opbg.net (E.V.F.); 6Società Italiana Ricerca Oli Essenziali, Viale Regina Elena 299, 00161 Roma, Italy

**Keywords:** biofilm modulation, *Pseudomonas aeruginosa*, cystic fibrosis, machine learning, essential oil

## Abstract

The opportunistic pathogen *Pseudomonas aeruginosa* is often involved in airway infections of cystic fibrosis (CF) patients. It persists in the hostile CF lung environment, inducing chronic infections due to the production of several virulence factors. In this regard, the ability to form a biofilm plays a pivotal role in CF airway colonization by *P. aeruginosa*. Bacterial virulence mitigation and bacterial cell adhesion hampering and/or biofilm reduced formation could represent a major target for the development of new therapeutic treatments for infection control. Essential oils (EOs) are being considered as a potential alternative in clinical settings for the prevention, treatment, and control of infections sustained by microbial biofilms. EOs are complex mixtures of different classes of organic compounds, usually used for the treatment of upper respiratory tract infections in traditional medicine. Recently, a wide series of EOs were investigated for their ability to modulate biofilm production by different pathogens comprising *S. aureus*, *S. epidermidis*, and *P. aeruginosa* strains. Machine learning (ML) algorithms were applied to develop classification models in order to suggest a possible antibiofilm action for each chemical component of the studied EOs. In the present study, we assessed the biofilm growth modulation exerted by 61 commercial EOs on a selected number of *P. aeruginosa* strains isolated from CF patients. Furthermore, ML has been used to shed light on the EO chemical components likely responsible for the positive or negative modulation of bacterial biofilm formation.

## 1. Introduction

The opportunistic pathogen *Pseudomonas aeruginosa* is a significant cause of healthcare-associated infections correlated with high morbidity and mortality in individuals with pneumonia, chronic obstructive pulmonary disease (COPD), or cystic fibrosis (CF) [1,2,3,4]. These infections are particularly problematic in intensive care units. For these reasons, this microorganism is included in the critical category of the World Health Organization’s (WHO) priority list of pathogens for which the discovery of new therapeutics is urgently needed [5]. *P. aeruginosa* can cause both acute and chronic infections, since its pathogenic profile originates from a large and variable arsenal of virulence factors and antibiotic resistance determinants. In the airways of CF patients, *P. aeruginosa* persists, inducing a chronic infection; furthermore, it is widely known that the CF pulmonary environment confers multiple advantages to *P. aeruginosa* over other pathogens, such as *Staphylococcus aureus* and *Klebsiella pneumoniae* [6]. The ability to form a biofilm plays a pivotal role in CF airway colonization by *P. aeruginosa*. Indeed, among its various virulence factors, the ability to produce highly structured biofilms confers important advantages, including phenotypic resistance to host defenses, antibiotics, and disinfectants [7]. These characteristics prevent bacterial clearance and allow the establishment of highly recalcitrant chronic infections [8,9].

A novel strategy to fight *P. aeruginosa* infection could derive from the identification of compounds acting on the biofilm phenotype without affecting bacterial vitality; these antibiofilm compounds could also enhance the effectiveness of conventional therapies, particularly in chronic infections such as CF [10,11].

Herbal antimicrobials are considered as a potential alternative in clinical settings for the prevention, treatment, and control of infections sustained by microbial biofilms [12]. Essential oils (EOs) are complex mixtures of different classes of organic compounds, and they are usually used for the treatment of upper respiratory tract infections in traditional medicine [13]. Furthermore, bacteria fail to develop resistance to multi-component treatments such as EOs due to their multitarget actions [14].

Recently, a wide series of EOs from Mediterranean plants were investigated for their ability to modulate biofilm production by different pathogens comprising *S. aureus*, *S. epidermidis,* and *P. aeruginosa* strains [15,16]. In this study, Machine learning (ML) algorithms were applied to develop classification models in order to suggest a possible antibiofilm action for each chemical component of the studied EOs. An analysis of the ML models indicated the chemical components possibly responsible for the inhibition or stimulation of bacterial biofilms. In two recent publications, ML-based clustering was used to develop a convergent microbiological protocol in which 61 EOs were evaluated on 40 clinical isolated of *S. aureus* and *P. aeruginosa* strains from CF patients [16,17]. First, the antimicrobial activity of each EO was tested against each *S. aureus* and *P. aeruginosa* clinical strain. Then, the antibiofilm activity was evaluated in the same *S. aureus* clinical isolates [17]. Based on these results, in the present study, we assessed the biofilm growth modulation exerted by the same EOs on a selected number of *P. aeruginosa* strains isolated from CF patients. Furthermore, ML has been used to shed light on the EO chemical components likely responsible for the positive or negative modulation of bacterial biofilm formation. 

## 2. Materials and Methods

### 2.1. Ethics Approval and Informed Consent

This research, performed according to the principles of the Helsinki Declaration, was approved by the ethics committee of the Children’s Hospital and Institute of Research Bambino Gesù (OPBG) in Rome, Italy (no. 1437_OPBG_2017 of July 2017). The individual participants and parents/legal guardians of the patients have signed an informed consent form included in the study. 

### 2.2. Description of P. aeruginosa Clinical Isolates from CF Patients 

Six representative clinical *P. aeruginosa* strains were used in this investigation, previously selected by a mean of unsupervised ML clusterization, as recently described [17]. 

Patients were treated according to the current standards of care [18]. Microbiological cultures were performed according to the approved guidelines as already described in Ragno et al. [17]. In Appendix A, the 18 qualitative descriptors used to cluster and define the six selected *P. aeruginosa* strains are described. Phenotypic and genotypic characteristics of these strains are summarized in Appendix A. The moderately virulent *P. aeruginosa* PAO1 (PAO1) and the highly virulent *P. aeruginosa* PA14 (PA14) were used as reference strains [19]. 

### 2.3. Biofilm Production Assay in the Presence of EO

The biofilm production was quantified in vitro by microtiter plate biofilm assay (MTP). A bacterial suspension (about 0.5 OD 600 nm) in the exponential growth phase was diluted into the wells of a sterile 96-well polystyrene flat base plate prefilled with medium containing or not containing each of the EOs listed in Appendix A, as previously reported [20]. Each EO was solubilized by adding DMSO, to generate a mother stock solution at 50% *v*/*v* concentration. As a control, the bacterial cells were grown in Brain Hearth Infusion broth (BHI, Oxoid, Basingstoke, UK) in the first row of the plate. In the second row the same culture medium was supplemented with each EO at a final concentration of 1.00% *v*/*v*. The incubation was performed aerobically overnight at 37 °C. After 18 h of incubation, planktonic cells were gently removed by washing each well three times with double-distilled water, and patted dry in an inverted position. Each well was stained with 0.1% crystal violet for 15 min at room temperature, rinsed twice with double-distilled water, and thoroughly dried to quantify the biofilm formation. The biofilm was subsequently solubilized with 20% (*v*/*v*) glacial acetic acid and 80% (*v*/*v*) ethanol. The total biomass of biofilm was spectrophotometrically quantified at 590 nm. Each data point is composed of four independent experiments, each performed in at least three replicates. 

### 2.4. Essential Oil Chemical Composition Analysis

The EOs are listed in Appendix A. They were purchased from Farmalabor srl (Assago, Italy) and their chemical composition was analyzed by gas chromatography-mass spectrometry (GC-MS). The adopted operative conditions followed Papa et al. [16]. Each component was identified by comparing the obtained mass spectra with those reported in the Nist 02 and Wiley mass spectra libraries. Linear retention indices (LRIs) of each compound were also calculated using a mixture of aliphatic hydrocarbons (C8–C30, Ultrasci Bologna, Bologna, Italy) injected directly into the GC injector. All analyses were repeated twice.

### 2.5. Machine Learning Binary Classification Modeling

All analysis were performed using the Python programming language (version 3.7, https://www.python.org/) [21,22] by executing in-house code in the Jupyter Notebook platform [16,17,20]. The chemical composition of each EO and the microbiological data were imported, subsequently loaded into a Python Pandas dataframe, and pre-processed to the final datasets to obtain the classification models. Scikit-learn (sklearn) [23] and the Pandas [24,25] libraries were used to implement Machine learning (ML) algorithm protocols. 

During model development, an unsupervised dimensionality reduction/transformation was performed with principal component analysis (PCA) [26] to extract 60%, 80%, 90%, and 100% of the explained variance (Appendix A). Different cut-off values related to the percentage of biofilm reduction/augmentation were used to develop ad hoc models to inspect strong, moderate, and weak biofilm inhibition and biofilm enhancement. In a departure from previous applications, a data augmentation (DA) approach was also implemented herein [27]. The EO dataset was augmented by means of composition random perturbation, while keeping the same bioactivity for each augmented related EO. In particular, for each EO, all the components were randomly modified by adding or subtracting up to 15% to/from each EO component, increasing the number of data rows by 10 (aug10) or 20 (aug20) times. In the case of unbalanced augmentation, for each EO, 10 new “virtual” records were generated (baug10 and baug20 in the table), while for the balanced process, with *w* being the weight of the EO class, it was augmented *w**10 times. Moreover, components represented by an occurrence of 2, 4, or 6 times were therefore eliminated from the training set. The robustness of the final models, as well as during the hyperparameters’ tuning, was evaluated by cross-validation (CV).

Due to the high number of considered hyperparameter combinations, the ML modeling strategy was conducted as follows:A first coarse ML model generation was run with 10 random hyperparameter combination runs from all possible considered combinations (Appendix A) [28];A second level of investigation was run with 100 random hyperparameter combination runs from all possible considered combinations (Appendix A) to select the optimal DA settings;A pre-final level was run with 1000 random hyperparameter combinations to check for protocol correctness, while extracting statistical coefficients for preliminary model evaluation;A final hyperparameter combination selection was performed by running 10,000 random combinations;The best model was finally further investigated with 1000 runs of DA perturbations, and the top scored model was used to deeply analyze the data.

Linear and non-linear ML classification algorithms were used to develop different models: random forest (rf), logistic regression (lr), support vector (sv), gradient bosting (gb), decision tree (dt), and *k* nearest neighbors (knn) as implemented in sklearn. The accuracy (ACC), F1 score, and Matthews correlation coefficient (MCC) were used to numerically and graphically evaluate the binary classification models. The importance of each chemical component present in EOs was independently evaluated through the “feature importance” (FI) and partial dependence (PD) [29] methods, as implemented in the Skater python library [30,31]. 

Models were validated by leave-some-out CV by means of five groups using the stratified K-fold method monitoring the average value of MCC obtained from 50 random CV iterations [15,32]. The selection of the final models was based on the MCC values.

## 3. Results

### 3.1. Biofilm Production Modulation by EOs 

The EOs’ ability to modulate *P. aeruginosa* biofilm production was evaluated at a concentration of 1.00 *v*/*v* % on the basis of a previous report [17]. The antimicrobial activity of the 61 EOs listed in Appendix A was evaluated, and the results are reported in Appendix A. Inactive EOs were investigated for their ability to modulate biofilm production. Biofilm production was compared to that of untreated bacteria (Table 1). 

### 3.2. Essential Oil Chemical Composition

The chemical compositions of the 61 EOs have already been reported as described in reference [16], and they are also reported in the Appendix A.

### 3.3. Machine Learning Models

#### 3.3.1. Datasets

Considering the antimicrobial activity data (Appendix A), the biofilm production investigations (Table 1), and the eight *P. aeruginosa* strains, a total of eight different initial datasets were loaded into a Pandas dataframe. Each dataset was composed of a data matrix of 61 rows (EO1–EO61, samples listed in Appendix A) and 240 columns (one bioactivity and 239 chemical components). To evaluate the underdevelopment of the ML model’s ability to discriminate between biofilm-inhibiting or biofilm-stimulating EOs, the biological data were binarized (partitioned into two classes) using different percentages of the biofilm production threshold value, SM. For all the strains used, threshold values of 40% (strong biofilm inhibition) and 120% (strong biofilm stimulation) were selected. 

For completeness, moderate biofilm inhibition (threshold of 80%) and a direct classification of biofilm inhibitors and enhancers (threshold of 100%) were also taken into consideration, and the results are reported in the Appendix A. As the antimicrobial data were too unbalanced, no tentative work was conducted in developing ML models.

#### 3.3.2. Classification Models

To avoid too many unbalanced datasets, the modeling was restricted to binarized data showing, at a maximum, a ratio of 10% ÷ 90% (or 90% ÷ 10%) data distribution, thus allowing the development of 27 models out of the 32 possible combinations (eight strains by four thresholds). 

Classification modeling at 40% and 120% thresholds were carried out with six different ML algorithms (rf, gb, sv, lr, dt and knn) using the introduced datasets. Initial classification models were built using the same protocol reported in reference [16], but, unfortunately, statistically acceptable models (MCC values greater than 0.4) were obtained only for two strain/threshold combinations (Appendix A). Similarly, only a few weak models were obtained for 80% and 100% threshold values (Appendix A). Recently, DA has been reported as a useful tool to develop ML models suffering from either an insufficient amount of data or the presence of noisy experimental data [33]. Despite the intrinsic power of ML, the latter conditions can lead to poor models as the available data do not cover the possible range of applications, such as EOs′ chemical composition variability. Therefore, DA was implemented herein in a new strategy to develop ML models (see Materials and Methods). Classification models were built with a number of latent variables corresponding to 60%, 80%, 90%, and 100% of the whole chemical components’ variance extracted by PCA. Moreover, to avoid the development of models driven by poorly represented components, those components with occurrences lower than 2, 4, and 6 were systematically removed from the training set. Hyperparameter optimization was carried out with a wide range of settings, leading from thousands to billions of combinations (Appendix A). Therefore, to speed up the calculations, a random search was used in place of the most common and exhaustive grid search. Random search hyperparamenters’ optimization was proved, having a probability of 95% of finding a combination of parameters within the optimal 5% with only 60 iterations [28]. Herein, the procedure described in the Material and Methods section led to the elaboration of more than three quarters of a million models (Appendix A) to seek the best combination of settings (DA and hyperparameters) to define eleven final ML models (Table 2). The initial DA and hyperparameter optimization was run with only 10 iterations and with coarse settings (Appendix A) leading to the generation of 2880 models for each of the 11 datasets of Table 2. For each dataset the top 3 models were selected leading to select the 33 preliminary ML models P1-P33 with cross-validated MCC values ranging from 0.34 to 0.78 (Appendix A). Then, the P1–P33 models were subjected to a further 100 iterations to select 11 models in which the DA settings (Appendix A) were finally selected, leading to the intermediate models I100_1–I100_11 characterized by MCC values in the 0.47–0.88 range (Appendix A). A third round of hyperparameter optimization was performed with 1000 random iterations while keeping the models’ I100_1–I100_11 DA settings, furnishing models I1000_1–I1000_11 (Appendix A) which were optimized to the pre-final ML models (PF1–PF27) through a further 10,000 random iterations. Interestingly, models PF1–PF11 were characterized by the same range MCC values of models I1000_1–I1000_11 and models I100_1–I100_11, thus indicating a sort of convergence being reached for the optimal hyperparameter selection (Appendix A). The models PF1–PF11 were then subjected to 100 rounds of iteration of random DA with the DA settings and hyperparameters selected using the associate models I100_1–I100_11 and PF1–PF11 themselves, respectively. The top-scoring DA final models F1–F11 were then selected, and the associated MCC, ACC, and F1 values calculated (Table 2). Models F1–F11 were finally analyzed through FI and PD values and plots to investigate the most important chemical components likely responsible for biofilm modulation (FIs) and to seek their statistical responsibility in each model. For completeness, the same procedures were applied using threshold values of 80% and 100% (Appendix A).

#### 3.3.3. Chemical Components Importance and Partial Dependences

Chemical component importance was evaluated through FIs and PDs. Each FI indicates a sort of absolute correlation coefficient for each of the chemical components (Appendix A), while the associated PD gives its negative, positive, or no influence. PDs’ positive or negative trends were evaluated through the Spearman correlation (SP) coefficient. The SP values were used to correct the corresponding FI into positive or negative weighted FIs (WFIs) and plotted. To reduce useless redundant values, only the top 10 and lowest 10 WFIs values were inspected (Figure 1 and Figure 2). The analysis of the WFI values led to the association of the overall effect on biofilm inhibition or stimulation for each chemical component (Table 3).

#### 3.3.4. Chemical Components Importance and Partial Dependences at 40% Biofilm Production Threshold Value

At a 40% biofilm production threshold value, good MCC, ACC, and F1 values were obtained for six out of the eight *P. aeruginosa* strains (models F1–F6, Table 2 and Figure 1). In particular, linalool, listed in the top 30 most frequent EOs’ components with a percentage of presence of about 60% (Appendix A), proved to be the chemical component most likely to be involved in strong biofilm production inhibition as identified in four out of six ML models (22P, 25P, 27P, 39P). Other compounds that seem to be important for a strong biofilm reduction are eucalyptol, linalyl anthranilate, geranyl acetate, bornyl acetate, cis-geraniol, sabinene, and cis-3-pinanone. Differently from linalool, these compounds are associated with the inhibition of biofilm production for one, two, or three strains. All together, the nine components might ensure a wide spectrum against the 22P, 25P, 27P, 37P, and 39P isolated strains. Interestingly, linalool and geranyl acetate are two of the most abundant components in EO54 and, in agreement with the above analysis, this EO showed a strong biofilm reduction with an average percentage of biofilm production as low as 31% against the 22P, 25P, 27P, 37P, and 39P isolated strains. Indeed, linalool was present at different percentages in seven of the eight more potent biofilm-reducing EOs (EO10, EO11, EO24, EO44, EO46, EO53, and EO54, each composition reported in Appendix A), combined mainly with eucalyptol and geranyl acetate, likely acting in a synergistic way. Interestingly β-caryophyllene, α-pinene, limonene, and p-cymene were indicated as important to decrease the biofilm production for different strains, while this had a negative impact on EOs’ biofilm inhibition for the other strains (Table 3). In contrast, β-pinene and carvacrol were found to exert only negative modulation on biofilm inhibition.

#### 3.3.5. Chemical Components Importance and Partial Dependences at a 120% Biofilm Production Threshold Value

As seen for the threshold value of 40%, at 120%, ML models (F6–F11, Table 2, and Figure 2) with MCC acceptable values were obtained for only five out of eight strains (PAO1, 25P, 26P, 27P, and 39P). Eucalyptol and o-cymene were the components calculated as likely to be responsible for slowing down biofilm production in PAO1, while thymol, p-cymene, citronellal, and carvacrol were mainly found as compounds possibly important for biofilm production stimulation. The balancing compounds for biofilm production enhancement were indicated to be linalool, linalyl anthranilate, limonene, and α-pinene. 

## 4. Discussion

Biofilm represents the strongest form of phenotypical resistance to the host immune defenses and antibacterial drugs operated by bacteria. It plays a pivotal role in the chronicization of many infections, including lung infections as in CF patients. The identification of new compounds able to interfere with biofilm development could lead to the removal of a primary cause of the persistence of infections.

In previous reports, it has been demonstrated that EOs can exert either antibacterial [15,17,34,35,36,37,38,39,40,41,42] or biofilm modulation effects [15,16,17,20,42,43,44,45,46,47,48]. As a continuation of a previously reported screen for antibacterial and antibiofilm EOs [15,16,17,20,42], herein, 61 previously investigated commercial samples have been evaluated for their abilities to modulate the biofilm formation of six *P. aeruginosa* clinical strains (22P, 25P, 26P, 27P, 37P, and 39P) in comparison with the reference strains PAO1 and PA14. Except for a few samples, the EOs tested at a concentration of 1.00% *v*/*v* showed a wide variability in either positively or negatively modulating bacterial biofilm production. A biofilm is continuously in equilibrium between accumulation and disruption, being subjected to a wide array of intracellular and extracellular factors. Therefore, it is not surprising that the same EO, that is a complex mixture of many chemical compounds (molecules), may act synergistically or anti-synergistically in stimulating or inhibiting biofilm development. The application of ML algorithms led to models that allowed the identification of the chemical compounds most related to strong biofilm growth inhibition. In particular, linalool (and to a lesser extent eucalyptol, linalyl anthranilate, geranyl acetate, bornyl acetate, cis-geraniol, sabinene, and cis-3-pinanone) is indicated as the most important component endowing EOs with a strong antibiofilm potency. In agreement with previous reports on several chemical constituents of the same EOs [16,17], it could be speculated that eucalyptol and linalool could be listed as common chemical compounds that reduce biofilms in both *S. aureus* and *P. aeruginosa* reference and clinical isolates strains. Indeed, Karuppia and coworkers, and Kifer and coworkers in two independent reports demonstrate that eucalyptol plays an antibiofilm role in *S. aureus* and *P. aeruginosa* [49,50], while linalool was independently pointed to by Lahiri and Kerekes as an important regulator of *S. aureus* and *P. aeruginosa* biofilm formation [51,52]. 

Regarding the biofilm enhancement driven by our 61 tested EOs, thymol, p-cymene, citronellal, and carvacrol were indicated by the ML models as those compounds important for biofilm production stimulation. In the face of our experimental evidence, a literature survey on Scopus (www.scopus.com, accessed on 1 March 2022) showed almost no reports on small molecules’ or EOs’ abilities to increase biofilm production. 

In this regard, 89 EOs extracted from Mediterranean plants previously screened for their biofilm modulation capability in *P. aeruginosa* PAO1 [15] and in four *Staphylococcus* strains [20] showed their abilities in stimulating biofilm production. The analysis of their composition by means of ML methods did highlight the important role of a few chemical compounds in modulating biofilm production. Nevertheless, the overall chemical compounds of the studied EOs were not overlapping with those investigated herein and therefore different conclusions were drawn. Interestingly, for sheer speculation, in previous published reports, limonene was indicated as a potential key molecule that, due to its lipophilic nature, could likely exert some gate role for different either anti-biofilm or pro-biofilm compounds. Herein, limonene and other hydrophobic components (α-pinene and p-cymene) seem to be confirmed to serve as enhancers (positively or negatively) for other components. 

In spite of reports supporting the above hypothesis on biofilm inhibition [12,49,50,51], further investigations on ad hoc selected EOs or their isolated chemical compounds are required to confirm the role of single molecules and their synergistic or anti-synergistic effects. 

In conclusion, in this study, according to previously published articles, the role of EOs and their chemical components is less obscure and ML algorithms have further confirmed their potential as valuable tools to shed light on EOs’ likely mechanism of activity. Furthermore, herein, the DA application proved to be a valid method to build robust models, when classical ML application failed. In particular, DA application seems particularly suitable for EOs, which are always critical for their scarce standardizability by chemists and medicinal chemists’ communities. As herein applied, the DA considers the composition variability of EOs obtained from the same plants, and also the intrinsic low ratio stability due the different and high volatility associated to each compound. 

## Figures and Tables

**Figure 1 microorganisms-10-00887-f001:**
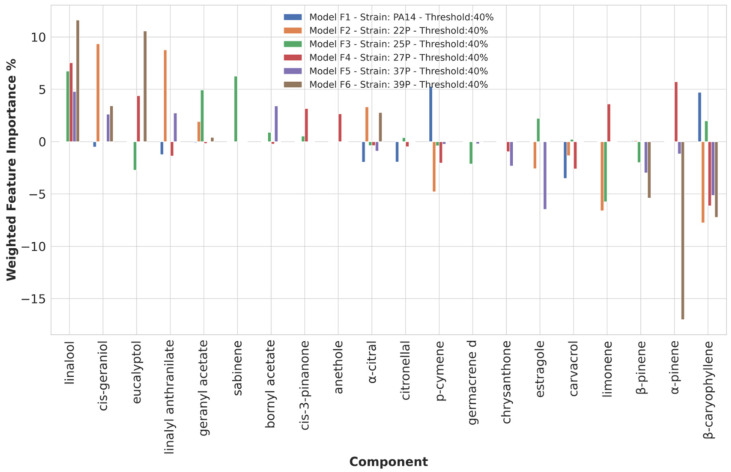
Weighted feature importance (WFI) plot for models F1 to F6 obtained on the dataset binarized at 40% biofilm inhibition. Positive bars are associated with inhibition of biofilm production, whereas negative bars are associated with augmented biofilm production. Only the 10 highest (anti-biofilm) and 10 lowest (pro-biofilm) values are displayed.

**Figure 2 microorganisms-10-00887-f002:**
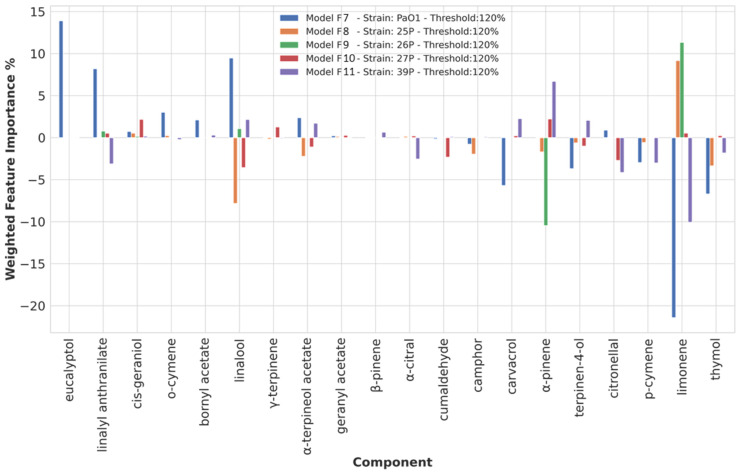
Weighted feature importance (WFI) plot for models F7 to F11 obtained on the dataset binarized at 120% biofilm inhibition. Positive bars are associated with inhibition of biofilm production, whereas negative bars are associated with augmented biofilm production. Only the 10 highest (anti-biofilm) and 10 lowest (pro-biofilm) values are displayed.

**Table 1 microorganisms-10-00887-t001:** Effect of EO on biofilm formation. Percentage of bacterial biofilm formation in the presence of each EO listed in Appendix A at a concentration of 1.00% *v*/*v* relative to untreated bacteria. Each data point is composed of four independent experiments, each performed with at least three replicates. NA: not applicable, being EO antimicrobial at tested concentration for this strain. At the concentration tested, the EO was antimicrobial, and consequently the biofilm modulation was not evaluated.

ID EOs	PaO1	PA14	22P	25P	26P	27P	37P	39P
EO1	171.68 ± 8.65	102.90 ± 5.14	87.26 ± 4.13	38.44 ± 1.92	191.93 ± 9.60	134.12 ± 6.71	137.52 ± 6.87	142.70 ± 7.13
EO2	244.23 ± 12.22	54.32 ± 3.26	101.82 ± 5.09	17.12 ± 1.03	113.99 ± 5.70	215.33 ± 12.92	97.38 ± 4.87	116.77 ± 5.84
EO3	143.37 ± 7.10	152.01 ± 7,60	NA	105.72 ± 5.29	79.10 ± 4.75	59.25 ± 2.96	NA	118.93 ± 5.95
EO4	183.16 ± 9.16	48.88 ± 8.65	NA	68.33 ± 3.41	193.18 ± 9.66	170.23 ± 10.21	NA	6.18 ± 0.37
EO5	80.92 ± 4.03	70.52 ± 4,23	89.54 ± 4.77	84.55 ± 5.07	66.84 ± 4.01	90.86 ± 4.54	0.59 ± 0.04	47.66 ± 2.86
EO6	133.04 ± 7.98	56.16 ± 2.80	86.25 ± 5.17	25.20 ± 1.51	97.21 ± 4.86	169.64 ± 8.48	77.81 ± 4.69	8.51 ± 0.51
EO7	119.72 ± 4.79	40.87 ± 2.39	93.03 ± 4.65	31.41 ± 1.89	82.52 ± 5.11	211.32 ± 12.68	80.80 ± 4.04	88.51 ± 4.42
EO8	130.36 ± 7.82	37.01 ± 2.22	97.07 ± 5.83	18.85 ± 0.94	63.22 ± 3.79	197.42 ± 9.87	76.18 ± 3.81	4.17 ± 0.21
EO9	90.15 ± 4.50	NA	NA	NA	NA	69.97 ± 3.50	NA	NA
EO10	83.80 ± 5.03	50.20 ± 2.51	84.81 ± 5.09	77.93 ± 3.90	81.86 ± 4.91	46.79 ± 2.33	0.57 ± 0.03	3.40 ± 0.17
EO11	161.28 ± 9.68	81.37 ± 4.88	26.42 ± 1.32	119.18 ± 5.96	45.07 ± 2.25	28.86 ± 1.44	26.21 ± 1.31	16.61 ± 0.83
EO12	200.22 ± 12.01	NA	84.57 ± 8.65	16.75 ± 0,83	77.79 ± 3.89	186.92 ± 11.21	85.37 ± 5.12	1.27 ± 0.08
EO13	150.64 ± 6.02	68.27 ± 3.41	30.90 ± 1.85	75.00 ± 3.75	30.53 ± 1.53	26.04 ± 1.30	50.39 ± 2.52	75.64 ± 3.78
EO14	NA	NA	152.27 ± 9.14	NA	107.26 ± 5.36	NA	NA	NA
EO15	66.96 ± 3.35	47.92 ± 2.39	32.98 ± 1.65	357.44 ± 17,87	33.60 ± 1.68	29.03 ± 1.45	48.47 ± 2.42	56.85 ± 2,84
EO16	91.84 ± 5.51	55.12 ± 3.31	100.85 ± 5.04	79.98 ± 4.80	69.69 ± 3,48	77.60 ± 4.65	0.18 ± 0.01	5.17 ± 0.31
EO17	174.80 ± 10.49	25.28 ± 1.26	86.63 ± 4.33	33.36 ± 1.67	83.67 ± 4.18	97.91 ± 4.89	98.82 ± 4.94	13.81 ± 0.83
EO18	94.21 ± 4.71	51.51 ± 3.09	88.52 ± 5.33	64.22 ± 3.85	56.53 ± 3.40	62.53 ± 9.60	0.33 ± 0.02	18.14 ± 0.91
EO19	96.59 ± 4.83	40.94 ± 2.05	83.87 ± 4.32	148.84 ± 7.48	70.91 ± 3.67	175.29 ± 7.40	68.80 ± 3.32	41.84 ± 1.78
EO20	101.78 ± 5.09	70.73 ± 3.54	81.14 ± 5.23	76.40 ± 3.75	97.94 ± 4.78	90.46 ± 4.78	0.99 ± 0.04	129.74 ± 6.12
EO21	121.96 ± 7.32	60.76 ± 3.04	94.28 ± 5.23	39.47 ± 3.85	86.12 ± 3.40	105.60 ± 5.40	219.63 ± 9.90	32.92 ± 1.76
EO22	NA	NA	NA	NA	NA	NA	NA	NA
EO23	91.76 ± 4.59	76.64 ± 4.61	85.30 ± 3.21	56.88 ± 3.00	65.40 ± 3.67	83.77 ± 4.41	93.96 ± 4.08	7.63 ± 3.40
EO24	43.68 ± 2.62	48.46 ± 2.42	61.60 ± 3.56	93.97 ± 5.78	119.89 ± 5.67	51.13 ± 3.40	0.72 ± 0.06	5.07 ± 2.21
EO25	80.63 ± 4,03	100.33 ± 5.02	62.94 ± 2.45	165.61 ± 8.98	95.18 ± 4.78	25.93 ± 1.78	91.99 ± 3.88	78.40 ± 3.87
EO26	81.08 ± 4.05	45.23 ± 2.26	280.83 ± 14.34	82.41 ± 3.96	63.88 ± 3.43	52.13 ± 2.61	0.89 ± 0.02	6.61 ± 3.21
EO27	38.50 ± 1.92	48.49 ± 2.91	60.12 ± 4.12	58.27 ± 2.65	143.64 ± 7.40	146.67 ± 7.89	0.34 ± 0.01	2.85 ± 0.65
EO28	69.88 ± 3,50	84.53 ± 5.07	83.90 ± 5.34	47.96 ± 2.21	84.26 ± 5.72	116.53 ± 5.12	101.67 ± 6.01	55.73 ± 3.01
EO29	99.57 ± 4.98	40.37 ± 2.42	68.14 ± 3.89	16.64 ± 0.69	79.53 ± 3.69	125.97 ± 5.67	89.27 ± 5.21	7.42 ± 3.43
EO30	101.11 ± 5.05	115.87 ± 5.79	79.12 ± 4.88	64.16 ± 3.44	90.40 ± 4.65	120.19 ± 6.71	100.04 ± 3.79	29.97 ± 1.21
EO31	91.35 ± 4.57	54.85 ± 2.74	116.72 ± 8.67	40.53 ± 2.01	154.39 ± 7.39	102.96 ± 5.12	106.85 ± 5.38	228.52 ± 9.91
EO32	NA	NA	NA	NA	NA	NA	NA	NA
EO33	78.70 ± 3.94	38.38 ± 1.92	244.16 ± 10.67	123.39 ± 6.43	122.63 ± 6.40	97.78 ± 4.78	80.55 ± 4.01	107.29 ± 5.21
EO34	58.60 ± 2.93	43.65 ± 2.19	59.81 ± 3.45	59.00 ± 2.67	76.95 ± 3.61	129.69 ± 6.12	0.52 ± 0.02	6.31 ± 3.01
EO35	99.57 ± 5.97	40.37 ± 2.02	68.14 ± 3.89	16.64 ± 0.53	79.53 ± 3.23	125.97 ± 6.28	89.27 ± 3.56	7.42 ± 3.74
EO36	57.65 ± 2.89	65.71 ± 3.29	74.02 ± 4.56	84.40 ± 4.21	87.98 ± 3.89	66.84 ± 3.46	0.96 ± 0.05	122.25 ± 6.02
EO37	150.48 ± 7.52	117.81 ± 5.80	NA	60.23 ± 3.03	141.57 ± 7.89	109.93 ± 4.99	NA	NA
EO38	149.62 ± 7.48	64.64 ± 3.23	22.61 ± 1.09	70.36 ± 3.43	48.92 ± 2.78	28.28 ± 1.98	NA	131.84 ± 5.89
EO39	NA	NA	NA	NA	NA	NA	NA	NA
EO40	122.50 ± 7.35	32.00 ± 1.60	80.43 ± 5.01	15.06 ± 0.43	67.86 ± 3.41	142.49 ± 6.42	72.49 ± 3.72	13.90 ± 0.99
EO41	141.92 ± 7.10	46.02 ± 2.31	14.36 ± 0.99	504.44 ± 19.95	128.47 ± 6	60.11 ± 3.11	98.48 ± 4.41	30.01 ± 1.23
EO42	86.65 ± 4.33	39.05 ± 1.78	84.45 ± 5.32	15.48 ± 0.32	77.26 ± 3.78	177.47 ± 8.91	88.80 ± 4.28	3.43 ± 0.18
EO43	127.46 ± 7.65	96.81 ± 4.34	14.11 ± 0.75	105.29 ± 5.42	95.01 ± 3.99	36.43 ± 1.21	71.83 ± 3.32	5.89 ± 0.78
EO44	71.79 ± 3.60	49.55 ± 2.53	15.25 ± 0.98	50.23 ± 2.32	111.58 ± 5.78	45.07 ± 2.13	88.43 ± 4.21	11.95 ± 0.98
EO45	148.56 ± 7.43	56.12 ± 2.76	23.45 ± 1.12	68.57 ± 3.79	50.81 ± 2.65	33.61 ± 1.21	38.15 ± 1.89	95.07 ± 4.01
EO46	147.21 ± 7.40	33.51 ± 1.78	15.90 ± 0.88	47.32 ± 2.01	103.80 ± 5.76	70.50 ± 3.56	NA	NA
EO47	58.99 ± 2.95	66.60 ± 3.45	22.08 ± 1.23	330.42 ± 14.54	24.85 ± 1.21	32.98 ± 1.67	33.64 ± 1.17	66.73 ± 3.23
EO48	304.94 ± 15.25	NA	50.99 ± 3.45	128.35 ± 7.89	244.63 ± 11.24	169.66 ± 7.47	11.06 ± 4.56	NA
EO49	78.66 ± 6.71	38.08 ± 1.92	142.03 ± 5.99	78.75 ± 3.65	94.61 ± 5.40	74.11 ± 3.33	131.36 ± 6.12	31.78 ± 1.12
EO50	103.14 ± 5.16	85.60 ± 4.76	12.10 ± 0.65	53.22 ± 3.01	122.85 ± 6.23	104.33 ± 5.78	94.66 ± 4.65	101.57 ± 5.62
EO51	102.62 ± 5.13	64.56 ± 3.76	14.08 ± 0.38	47.79 ± 2.94	104.82 ± 5.24	52.98 ± 1.98	82.85 ± 4.10	4.75 ± 0.28
EO52	113.76 ± 6.82	70.07 ± 3.56	71.95 ± 3.89	21.66 ± 1.09	172.35 ± 7.91	193.17 ± 6.78	87.94 ± 4.21	1.85 ± 0.09
EO53	134.51 ± 6.71	58.00 ± 2.58	15.85 ± 0.94	46.60 ± 2.45	134.77 ± 35.76	56.71 ± 2.12	76.66 ± 3.56	5.33 ± 0.27
EO54	100.73 ± 5.04	50.85 ± 2.27	17.21 ± 1.52	47.43 ± 2.76	106.48 ± 4.91	58.51 ± 3.23	25.76 ± 1.21	6.26 ± 0.82
EO55	118.70 ± 5.93	66.92 ± 3.34	144.27 ± 7.98	34.77 ± 1.96	86.90 ± 4.20	58.66 ± 3.01	118.57 ± 5.67	60.23 ± 3.21
EO56	90.11 ± 4.50	71.63 ± 6.76	110.29 ± 6.89	45.86 ± 2.91	91.28 ± 4.61	87.53 ± 3.78	163.12 ± 8.54	66.06 ± 3.21
EO57	55.72 ± 3.34	69.00 ± 4.65	15.78 ± 0.54	47.85 ± 2.04	190.73 ± 9.67	110.55 ± 6.11	92.72 ± 3.78	88.24 ± 3.79
EO58	76.24 ± 3.81	37.82 ± 2.18	21.54 ± 1.33	153.50 ± 6.72	78.89 ± 3.46	34.90 ± 1.67	54.29 ± 2.21	102.60 ± 3.89
EO59	160.70 ± 8.03	44.94 ± 2.56	17.32 ± 0.77	46.50 ± 2.78	166.22 ± 5.89	106.13 ± 5.43	NA	NA
EO60	232.46 ± 11.62	137.74 ± 7.53	82.93 ± 4.67	23.82 ± 1.21	284.60 ± 19.11	280.46 ± 7.12	NA	NA
EO61	352.32 ± 17.67	652.82 ± 38.65	105.04 ± 5.33	67.31 ± 3.49	627.58 ± 28.12	210.31 ± 8.98	58.53 ± 3.40	NA
Number Values	57	54	54	56	57	57	49	50

**Table 2 microorganisms-10-00887-t002:** Optimized final models obtained with 100 random iterations of data augmentation.

Model	Strain	Threshold	ML Method	MCC		ACC		F1	
				Fit	CV	Fit	CV	Fit	CV
F1	PA14	40	gb	1.00	0.62	1.00	0.91	1.00	0.67
F2	22P		dt	0.66	0.11	0.85	0.69	0.69	0.19
F3	25P		gb	1.00	0.47	1.00	0.80	1.00	0.59
F4	27P		gb	1.00	0.66	1.00	0.91	1.00	0.71
F5	37P		gb	0.71	0.71	0.88	0.88	0.80	0.80
F6	39P		rf	1.00	0.59	1.00	0.80	1.00	0.82
F7	PAO1	120	svm	0.41	0.56	0.72	0.79	0.77	0.84
F8	25P		dt	0.78	0.35	0.93	0.79	0.96	0.87
F9	26P		dt	1.00	0.64	1.00	0.86	1.00	0.90
F10	27P		svm	0.50	0.64	0.79	0.84	0.85	0.89
F11	PAO1		svm	1.00	0.88	1.00	0.98	1.00	0.99

**Table 3 microorganisms-10-00887-t003:** Summary on predicted essential oils’ (EO) single compounds influence on biofilm modulation at 40% and 120% threshold values. Compounds with an absolute value of FWI higher than 3 are listed.

Threshold	Anti-Biofilm	Strains	Pro-Biofilm	Strains
40	linalool	22P, 25P, 27P, 39P		
	eucalyptol	27P, 39P		
	linalyl anthranilate	22P, 37P		
	geranyl acetate	22P, 25P		
	bornyl acetate	37P		
	cis-3-pinanone	27P		
	cis-geraniol	22P, 37P, 39P		
	sabinene	25P		
	β-caryophyllene	PA14, 25P	β-caryophyllene	22P, 27P, 37P, 39P
	α-pinene	27P	α-pinene	39P
			β-pinene	39P
	limonene	27P	limonene	22P, 25P
			carvacrol	PA14
	p-cymene	PA14	p-cymene	22P
120	eucalyptol	PAO1		
	linalyl anthranilate	PAO1	linalyl anthranilate	37P
	o-cymene	PAO1		
	linalool	PAO1	linalool	25P, 27P
			thymol	PAO1, 25P
	limonene	25P, 26P	limonene	PAO1, 37P
			p-cymene	PAO1, 37P
			citronellal	37P
			terpinen-4-ol	PAO1
	α-pinene	37P	α-pinene	26P
			carvacrol	PAO1

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
