# Peer review of "Essential Oils Biofilm Modulation Activity and Machine Learning Analysis on Pseudomonas aeruginosa Isolates from Cystic Fibrosis Patients"

_microorganisms, 2022, doi:10.3390/microorganisms10050887_

Round 1

Reviewer 1 Report

Authors present experimental data as well as computational analysis using machine learning (ML)  algorithms for a mixtures of essential oils (EO) in application to P. aeruginosa biofilm erradication.  Practically, authors focussed on search for single EO responsible for biological activity. The obtained results indicate general usefullness of such a method for solving such complex problems. However,  methods still require fine refinement. Also it would be nice to see them more general.

Article is interesting. However, it requires some corrections.  For example, it is not clear for the reader which experimental data (have been published before by the same group and which are new.  It would be good to know from the text how many EO's are present in every commercial sample. More description in Table's headings are necessary; e.g. Table 1, what is NA and why it is applicable to all strains studied? May be EO22 and 32 should be removed from the set?  What are units? How learning set was prepared?

Major problem is that single component can be antimicrobially/antybiofilm active on one type of bacteria (funghi) and not active on another one, even within the same species.  Moreover, EO's can express synergy or anti-synergy with other components of the mixture. Nevertheless, relatively robust statistically significant ML models were developed, and analyzed in terms of component importance and partial relationship that led to identification of chemical components mainly responsible for biofilm production or inhibition. Feature that is not considered is in such an approach is that identification of the  major essential oils (chemical structure) responsible for expressing bioactivity might be not enough. For example, it would be interesting if basic set might be supplemented with assignment of the EO's mechanism of action. This would allow to categorize single EO's in groups, different that those already prepared for the present calculations. Particularly that mechanisms  are already known for many EO's. 

Summing up, manuscript might be published with minor revision.

Author Response

Reply to reviewer 1

Reviewer 2 Report

In manuscript "Essential oils biofilm modulation activity and machine learning analysis on Pseudomonas aeruginosa isolates from cystic fibrosis patients'" Machine Learning (ML) algorithms were applied in order to suggest a possible antibiofilm action for each chemical component of the studied EOs.
My main objection concerns the performance of anti-adhesion experiments or the inhibition of biofilm formation experiments.
I don't think it's necessary to use the term opportunistic bacterial culture, it's enough to mention bacterial suspension.
Was the original EO used in the experiments or it was diluted in a solvent such as DMSO?
If such a low concentration of EO was used directly, can you be sure that it has dissolved in the used medium and that it has shown its antibiofilm effect?

Although Pseudomonas is known for its resistance to various antimicrobial agents, the results shown in Table SM8 do not show encouraging results and a small number of EOs show inhibition of biofilm formation.
The title of the table is confusing and it is not clear to me whether you are showing biofilm inhibition or antimicrobial activity.
Why didn't you use higher EO concentrations to get more data for analysis?
Have you tested the individual performance of the main components of EO or has everything been done by mathematical modeling?
How do you explain the fact that certain EOs led to enhanced biofilm production shown in Table 1?

What was used as a control in the implementation of experiment 2.3.?
Can it be concluded which EOs showed the best antibiofilm effect or is the only goal of manuscript to show mathematical modeling?

Author Response

Reply to reviewer 2

Round 2

Reviewer 2 Report

The authors have explained all the ambiguities and I have no further comments.